# Probiotics attenuate valproate-induced liver steatosis and oxidative stress in mice

**Wenfang Song[1], Xinrui Yan[1], Yu Zhai[1], Jing Ren[1], Ting Wu[1], Han Guo[1], Yu Song[2], Xiaojiao Li[3], Yingjie Guo[1,4]***

1 School of Life Sciences, Jilin University, Changchun, China, 2 Key Laboratory of Utilization and Conservation for Tropical Marine Bioresources, Ministry of Education, Key Laboratory for Protection and Utilization of Tropical Marine Fishery Resources, College of Fishery and Life Science, Hainan Tropical Ocean University, Sanya, China, 3 Phase I Clinical Trial Center, The First Hospital of Jilin University, Changchun, China, 4 National Engineering Laboratory of AIDS Vaccine, Jilin University, Changchun, China

* guoyingjie@jlu.edu.cn

**Data Availability Statement:** All relevant data are within the manuscript and its Supporting Information files.

**Funding:** The authors acknowledge financial support from National Natural Science Foundation

## Abstract

Valproate (valproic acid, VPA), a drug for the treatment of epilepsy and bipolar disorder, causes liver steatosis with enhanced oxidative stress. Accumulating evidences exhibit that gut microbiota plays an important role in progression of nonalcoholic fatty liver disease (NAFLD). However, whether gut microbiota contributes to VPA-caused hepatic steatosis needs to be elucidated. A mixture of five probiotics was selected to investigate their effects on liver steatosis and oxidative stress in mice orally administered VPA for 30 days. Probiotics treatment significantly attenuated the hepatic lipid accumulation in VPA-treated mice via inhibiting the expression of cluster of differentiation 36 (CD36) and distinct diacylglycerol acyltransferase 2 (DGAT2). Meanwhile, probiotics exerted a protective effect against VPA-induced oxidative stress by decreasing the pro-oxidant cytochrome P450 2E1 (CYP2E1) level and activating the Nrf2/antioxidant enzyme pathway. Moreover, VPA treatment altered the relative abundance of gut microbiota at the phylum, family and genera levels, while probiotics partially restored these changes. Spearman's correlation analysis showed that several specific genera and family were significantly correlated with liver steatosis and oxidative stress-related indicators. These results suggest that probiotics exert their health benefits in the abrogation of liver steatosis and oxidative stress in VPA-treated mice by manipulating the microbial homeostasis.

## Introduction

Valproate (valproic acid, VPA) is widely prescribed to treat epilepsy and bipolar disorder in several decades. However, long-term treatment with VPA causes different degrees of hepatotoxicity from steatosis to acute liver failure [1]. Liver steatosis is the most common side effect, which occurs in 61% patients receiving VPA for more than two years [2]. If no specific treatment, liver steatosis may progress into more severe live injury such as nonalcoholic steatohepatitis, fibrosis, cirrhosis, and hepatocellular carcinoma.

of China (Project: 82104314), Natural Science Foundation of Jilin Province (Project: YDZJ202301ZYTS034). The funders had no role in study design, data collection and analysis, decision to publish, or preparation of the manuscript.

**Competing interests:** The authors have declared that no competing interests exist.

A number of studies have demonstrated that there is a close association between oxidative stress and VPA-caused liver steatosis [3–5]. VPA can enhance hepatic oxidative stress by inhibiting mitochondrial fatty acid β-oxidation or impairing ROS detoxification system [6, 7]. Our recent study found that CYP2E1 was also responsible for ROS accumulation upon VPA treatment, which Further promoted CD36-mediated lipid uptake and DGAT2-mediated triacylglycerol synthesis, causing hepatic lipid accumulation [8]. It indicates that oxidative stress contributes to VPA-caused liver steatosis.

Interestingly, serveral recent studies found that VPA administration markedly altered the composition and structure of gut microiota in both epileptic patients and rats [9, 10]. A consecutive 28-day treatment with VPA to rats significantly changed the bacterial richness and diversity with an increase in relative abundance of *Actinobacteria*, *Firmicutes* and a decrease in relative abundance of *Bacteroidetes* at the phylum level [9]. Meanwhile, in epileptic patient receiving VPA treatment, some gut microbiota were correlated with the alteration of liver function-related biochemical parameters including alanine aminotransferase (ALT), aspartate aminotransferase (AST), triglyceride (TG), cholesterol (TC) and *lactate* dehydrogenase (LDH) [10]. These findings suggest that there mignt be a link between gut microbiota dysbiosis and VPA-caused metabolic disorder.

Accumulating evidences point out that gut microbiota plays a critical role in the occurance and development of nonalcoholic fatty liver disease (NAFLD) [11]. Patients and animals with NAFLD had the altered composition and structure of gut microbiota at different levels [12]. Additionally, gut microbiota seems to be directly or indirectly associated with the change of redox state in the intestine or other tissue including liver [13]. Therefore, we suppose that gut microbiota modulation may be *beneficial* for improving hepatic oxidative stress and lipid metabolism in VPA-treated mice.

Probiotics, alive microorganisms with health benefits, have been demonstrated to exert beneficial roles in the restoration of microbial homeostasis and abrogation of NAFLD [14]. The administration of probiotics may provide a new strategy in the prevention or treatment of liver steatosis. Probiotics such as *Lactobacillus*, *Bifidobacterium* and *Streptococcus* had positive effects on the attenuation of NAFLD [12]. Especially, *Lactobacillus* and *Bifidobacterium* exhibited excellent antioxidative activities against oxidative stress-related diseases [15, 16]. Here we selected a mixture of five probiotics including *Lactobacillus plantarum* LP45, *Lactobacillus paracasei* YMC1069, *Bifidobacterium longum* L693, *Bifidobacterium bifidum* TMC3115 and *Streptococcus thermophiles* S131, and assessed their impacts on VPA-caused oxidative stress and liver steatosis in mice.

## Materials and methods

### Chemicals and reagents

Valproate was purchased from Sigma-Aldrich (St louis, MO, USA). *Lactobacillus plantarum* LP45, *Lactobacillus paracasei* YMC1069, *Bifidobacterium longum* L693, *Bifidobacterium bifidum* TMC3115 and *Streptococcus thermophiles* S131 were gifts from Hebei Yiran Biotechnology Co., Ltd.

### Animals and treatment

Male C57B/6 J mice at 6 weeks of age (20–22 g) were obtained from Liaoning Changsheng biotechnology Co., Ltd (Benxi, China). All the *mice* had *free* access to *food and water* and housed in a specific pathogen-free cage at $25 \pm 2°C$, $55 \pm 15\%$ humidity, and 12-hour alternating light-dark cycle circumstance. Mice were randomly divided into control group, VPA group, probiotics group, and VPA plus probiotics group (n = 6 each group) and fed with vehicle, VPA (500

mg/kg), a mixture of five probiotics ($3 \times 10^8$ CFU *Lactobacillus plantarum* LP45, $3 \times 10^8$ CFU *Lactobacillus paracasei* YMC1069, $3 \times 10^8$ CFU *Bifidobacterium longum* L693, $3 \times 10^8$ CFU *Bifidobacterium bifidum* TMC3115, and $0.5 \times 10^8$ CFU *Streptococcus thermophiles* S131), and VPA plus probiotic mixture for consecutive 30 days, respectively.

**Health and behavior of mice were monitored every day.** Body weights were measured each three days. Unfortunately, one mouse in the VPA plus probiotics group accidently died during the oral administration. All the mice were killed by $CO_2$ euthanasia 24 hours after the last administration with all efforts to minimize mice' suffering, then serum, liver tissues or feces were harvested for further analysis. All animal protocols and experiments were approved by the Animal Ethics Committee of Jilin University, in accordance with the Guidelines for the Care and Use of Laboratory Animals of Jilin University (Protocol Number: 2021-PZ012). Everyone obtained informed written consent. All operators have received the qualifications of animal experiments and professional training in animal welfare ethics in the School of Life Sciences of Jilin University, and proficient in the relevant operating skills of this experiment.

## Biochemical analysis

The TC, TG, ALT and AST levels in liver or serum were analyzed by commercial assay kits (Jiancheng Bioengineering Institute, Nanjing, China). The malondialdehyde (MDA) and glutathione (GSH) levels in liver tissues were analyzed using commercial assay kits (Solarbio, Beijing, China).

## Histopathological examination and immunostaining

Fresh liver tissues were fixed in 10% formalin and embedded in paraffin. Sections at 5 μm thick were used for hematoxylin and eosin (H&E) staining and 4-hydroxy-2-nonenal (4-HNE) immunostaining. The frozen liver tissues were fixed in optimum cutting temperature (O.C.T) compound. Sections at 8 μm thick were used for Oil-red O staining. Images were collected by a microscope (IX70, Olympus, Japan) at 200-fold magnification.

## qRT-PCR analysis

Total RNA was extracted from the liver of mice by a Total RNA Kit (OMEGA, Norcross, GA, USA). cDNA was synthesized from 2 μg of total RNA using the PrimeScriptTM reverse transcription reagent kit (TAKARA, Beijing, China). qRT-PCR was performed on a LightCycler®480 System (Roche, Switzerland) using SYBR green PCR Master Mix (TAKARA, Beijing, China), cDNA template, and gene-specific primers. Primer sequences for *CYP2E1*, *FATP2*, *FATP5*, *CD36*, *FAS*, *SCD1*, *ACC*, *DGAT1*, *DGAT2*, *SREBP-1c*, *PXR*, *PPARα*, *PPARγ*, and *GAPDH* were listed previously [8]. Primer sequences for *HO-1*, *Nrf2*, *SOD*, *CAT*, *GST*, *HMGCS*, *HMGCR* and *LXR-α* were shown in S1 Table. The comparative $C_t$ ($\Delta\Delta C_t$) method was used to quantitate the relative mRNA level.

## Western blot analysis

The total and nuclear protein in liver tissues were extracted using RIPA buffer containing 1 mM PMSF (Beyotime, Shanghai, China) and nuclear protein extraction kit (Solarbio, Beijing, China), respectively. Extracted proteins were separated on SDS-PAGE gels, transferred to PVDF membranes, incubated with anti-HO-1, anti-β-actin antibodies (BBI, Shanghai, China), anti-Nrf2, anti-Histone H3, anti-CD36 antibodies (Proteintech, Rosemount, IL, USA), anti-CYP2E1, anti-DGAT2 antibodies (Abcam, Cambridge, MA, USA), then visualized by the Odyssey Infrared Imaging System (Li-Cor). Densitometric quantification of protein bands was

performed using Odyssey V3.0 (Li-Cor) normalized to Histone H3 or β-actin and calculated as the fold-change compared with the control group.

## 16s rRNA sequencing analysis

DNA was extracted from the feces of mice using a stool DNA isolation Kit (TianGen, China). Universal primers F343 (5′-TACGGRAGGCAGCAG-3′) and R789 (5′-AGGGTATVTAATCCT-3′) were used to amplify the V3-V4 regions of the bacterial 16S rRNA gene. Sequencing was performed on an Illumina MiSeq platform at Shanghai OE Biotech. Co., Ltd. After high quality reads obtained from quality filtering, the normalized operational taxonomic unit (OTU) tables were generated from chimeras removal of the raw data for diversity and statistical analyses.

## Statistical analysis

Data were displayed as mean ± standard deviation. To determine the effect of VPA on the mRNA levels of some genes involved in FA uptake, FA synthesis, TG synthesis, TC metabolism and nuclear receptor, the comparision between control and VPA group was performed using the Student's $t$ test. For analyzing the body weight, comparisons between time-based measurements within each group were performed with analysis of variance for repeated measurement. For the other experimental studies including control, VPA, probiotics and VPA plus probiotics groups, one-way analysis of variance (ANOVA) with Tukey's multiple comparison test was adopted. The Spearman's rank correlation coefficients between the relative abundance of gut microbiota and liver steatosis and oxidative stress-related indicators were determined for correlational statistical analysis. All analyses were accomplished using SPSS 20.0 (IBM SPSS), and a $P$ value less than 0.05 was considered statistically significant.

## Results

### Probiotics attenuated VPA-caused liver steatosis and oxidative stress

Previous studies have demonstrated that consecutive 30-day treatment with VPA to mice caused liver steatosis [8]. Considering a close relationship between gut microbiota and NAFLD, we observed the effects of probiotics on VPA-caused liver steatosis. As shown in Fig 1A, treatment with VPA alone or in combination with probiotics significantly decreased the body weight of mice throughout the 30 days, as compared to vehicle treatment, but had no significant influences on liver weight and liver/body weight ration at the end of the experiment. Interestingly, probiotics dramatically inhibited the enhanced TC and TG contents or oil red O staining in the liver of mice after VPA treatment, indicating the decreased liver lipid accumulation by probiotics (Fig 1B). Meanwhile, the increased hepatic oxidative stress in VPA-treated mice as detected by the reduced GSH level, the elevated MDA level, and the enhanced 4-HNE immunostaining was also significantly alleviated by probiotics (Fig 1C). The H&E staining indicated the attenuated VPA-caused hepatic microvesicular steatosis in mice after co-treatment with probiotics, but the serum ALT and AST levels were comparable among the different groups (Fig 1D).

### Probiotics decreased oxidative stress by inhibiting the pro-oxidant CYP2E1 level and activating the Nrf2/antioxidant enzyme pathway

To clarify the molecular mechanism by which probiotics decreased the oxidative stress in VPA-treated mice, we first explored the level of CYP2E1, which has been demonstrated to promote ROS production and liver steatosis in VPA-treated mice [8]. As expected, VPA treatment

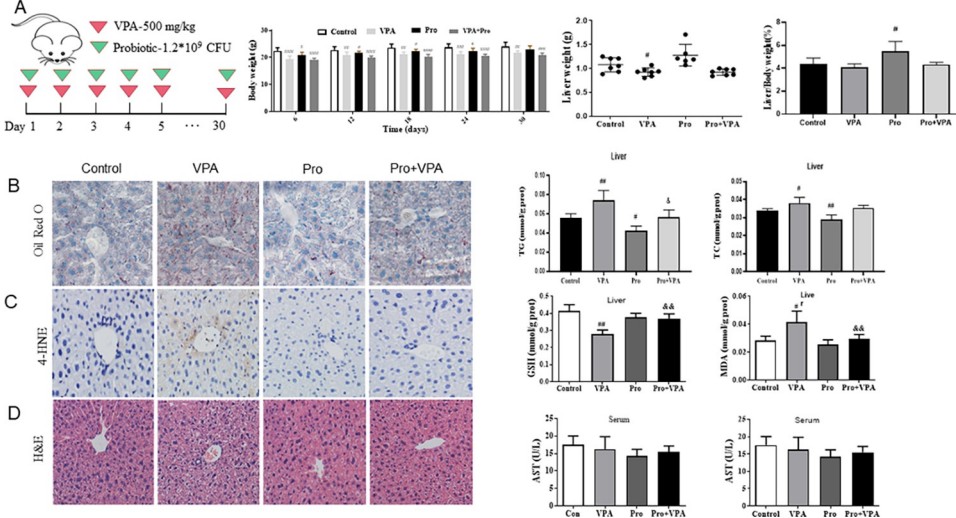

**Fig 1. Effect of probiotics on liver steatosis and oxidative stress in VPA-treated mice.** (A) Body weight, liver weight and liver/body weight ratio of mice; (B) Oil Red O staining (200x magnification), TG and TC levels in liver; (C) 4-HNE immunostaining (200x magnification), GSH and MDA levels in liver; (D) H&E staining (200x magnification), ALT and AST levels in serum. Data were expressed as the MEAN±SD.#$p < 0.05$, ##$p < 0.01$, ###$p < 0.001$ versus control group, &$P < 0.05$, &&$P < 0.01$, &&&$P < 0.001$ versus VPA group.

significantly elevated the mRNA and protein level of hepatic CYP2E1 (Fig 2A and 2B). Interestingly, probiotics treatment significantly attenuated the enhanced CYP2E1 level. Further, the levels of several antioxidant enzymes including CAT, GST, SOD, and HO-1 were evaluated. VPA treatment markedly decreased the mRNA levels or enzymatic activities of SOD, CAT or GST in liver or serum, while probiotics treatment alleviated these changes to different degrees

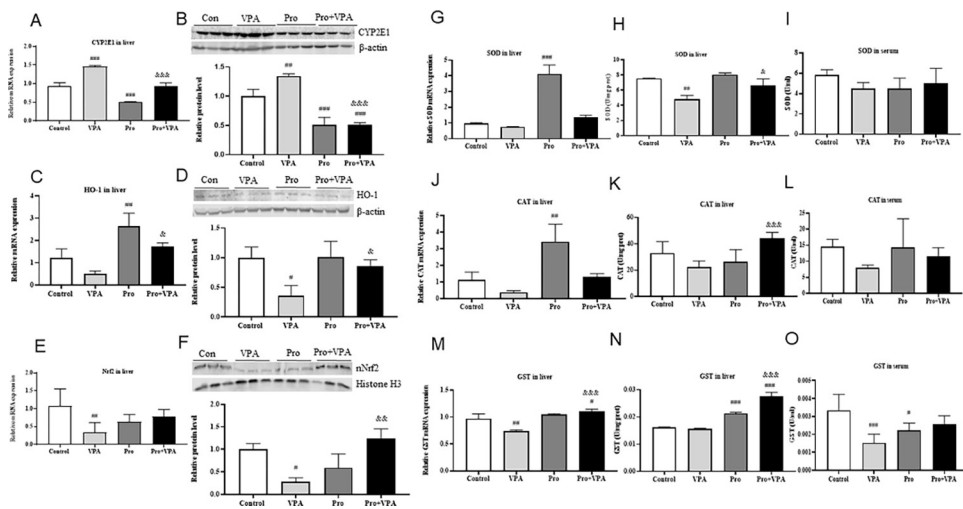

**Fig 2. Effect of probiotics on the levels of CYP2E1 and antioxidant enzymes in VPA-treated mice.** (A) mRNA and (B) protein levels of CYP2E1 in liver; (C) mRNA and (D) protein levels of HO-1 in liver; (E) mRNA and (F) nuclear protein levels of Nrf2 (nNrf2) in liver; (G) Hepatic mRNA level, (H) hepatic enzymatic activities and (I) serum enzymatic activity of SOD; (J) Hepatic mRNA level, (K) hepatic enzymatic activities and (L) serum enzymatic activity of CAT; (M) Hepatic mRNA level, (N) hepatic enzymatic activities and (O) serum enzymatic activity of GST. Data were expressed as the MEAN±SD. #$p < 0.05$, ##$p < 0.01$,###$p < 0.001$ versus control group, &$P < 0.05$, &&$P < 0.01$, &&&$P < 0.001$versus VPA group.

(Fig 2G–2O). Moreover, the decreased mRNA and protein levels of HO-1 in VPA group were also significantly prevented by probiotics (Fig 2C and 2D), similar with the alteration of mRNA and nuclear protein levels of Nrf2 (Fig 2E and 2F).

## Probiotics inhibited lipid accumulation by downregulating the CD36 and DGAT2 levels

To explore the underlying mechanism of VPA-caused lipid accumulation, the mRNA levels of some genes involved in fatty acid (FA) uptake, FA synthesis, TG synthesis, TC metabolism and nuclear receptor were analyzed by qRT-PCR method (Fig 3A). Among these selected genes, the expression of CD36 and DGAT2 was significantly upregulated upon VPA treatment, indicating the enhanced uptake of fatty acid and conversion from diglycerides into triglycerides in VPA-treated mice. Interestingly, the elevated mRNA and protein levels of both CD36 and DGAT2 were markedly inhibited by probiotics (Fig 3B and 3C). It suggests that probiotics could inhibit VPA-induced lipid accumulation via the downregulation of CD36 and DGAT2.

## Probiotics recovered gut microbial community structure

The 16S rRNA gene sequencing in fecal samples was accomplished to explore the effect of probiotics on gut microbiota composition. Microbial β diversity analysis based on PCoA plot showed a clear separation of bacterial composition between VPA group and control group, while VPA plus probiotics group was clustered between them (Fig 4C). Consistently, microbial

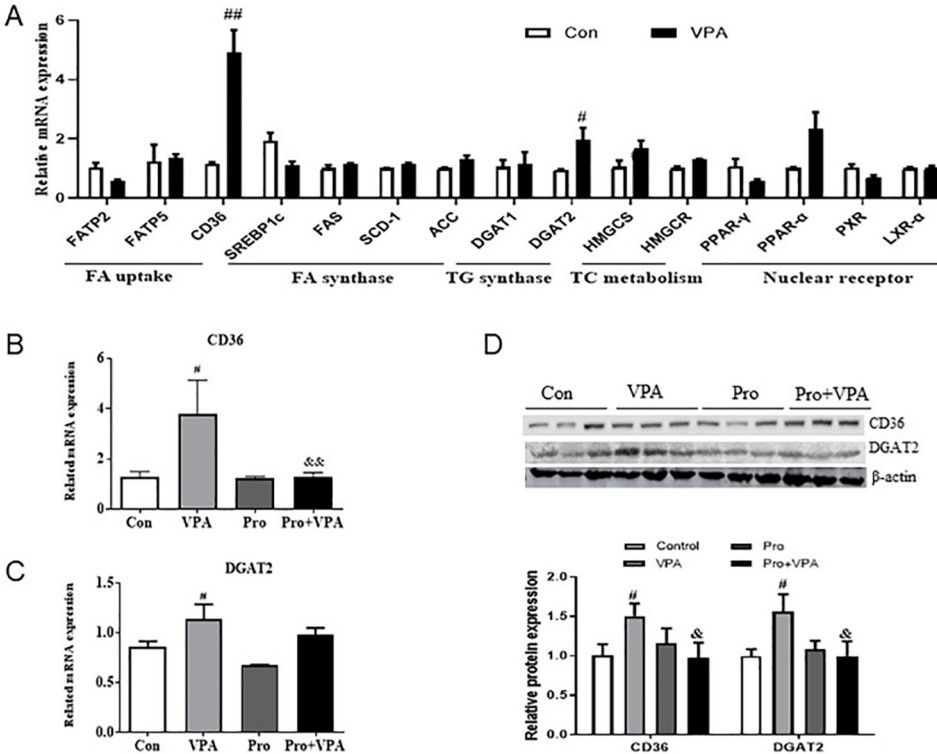

**Fig 3. Effect of probiotics on the expression of CD36 and DGAT2 in VPA-treated mice.** (A) The mRNA expression of genes involved in fatty acid (FA) uptake, FA synthesis, TG synthesis, TC metabolism and nuclear receptor in liver; (B) CD36 and DGAT2 mRNA levels in liver; (C) Protein levels of CD36 and DGAT2 in liver. Data were expressed as the MEAN±SD. $^{#}p < 0.05$, $^{##}p < 0.01$ versus control group, $^{&}P < 0.05$, $^{&&}P < 0.01$ versus VPA group.

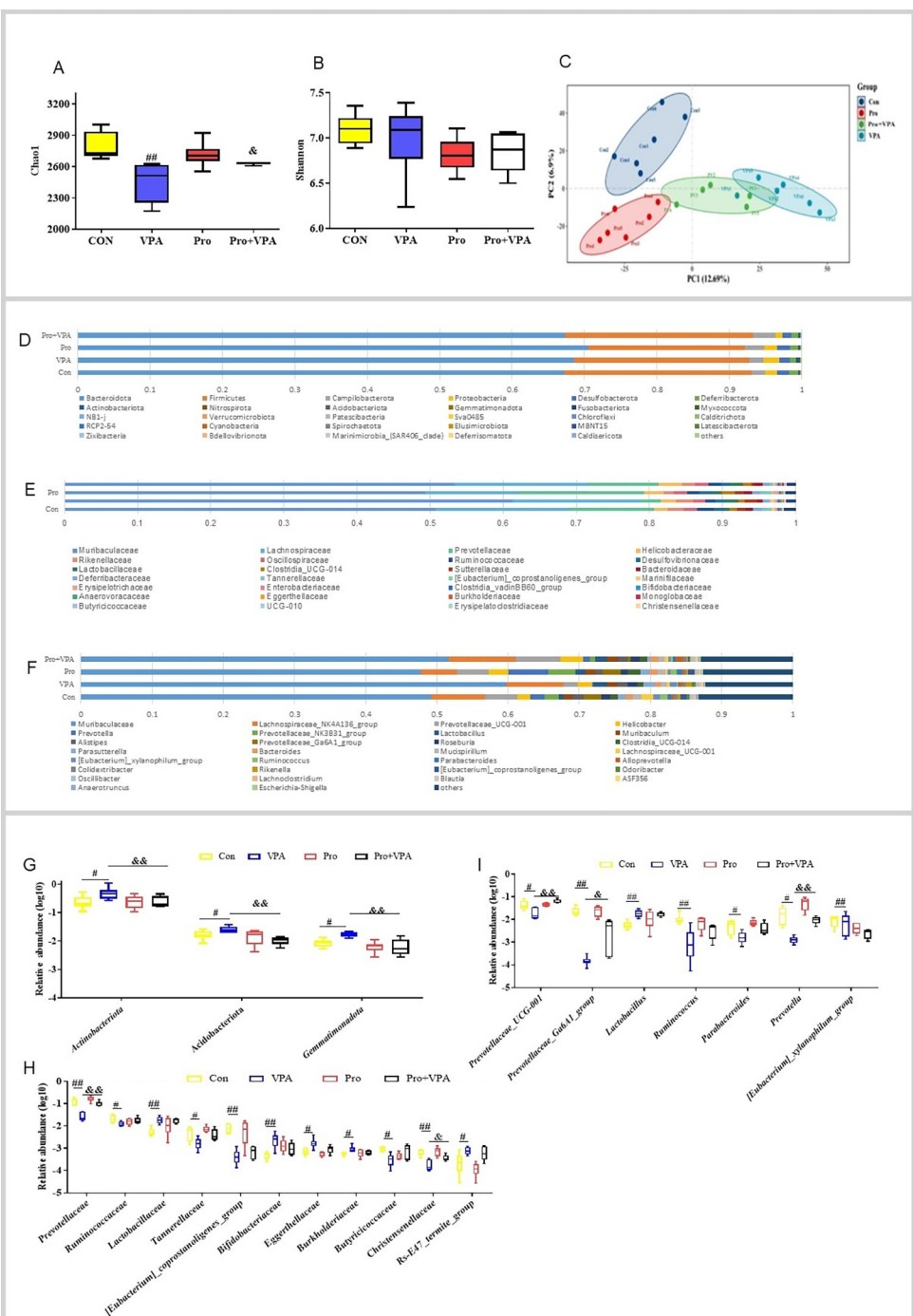

**Fig 4. Effect of probiotics on gut microbiota richness and composition in VPA-treated mice.** 16S rRNA sequencing analysis of isolated fecal DNA across the different groups. (A) Chao1 index; (B) Shannon index; (C) Principal co-ordinates analysis (PCoA); The relative abundance of the top30 taxa at the (D) phylum, (E) family, and (F) genus levels; The relative bacterial abundance with significant difference between two groups at the (G) phylum, (H) family, and (I) genus levels. Data were expressed as the MEAN±SD. #$p < 0.05$, ##$p < 0.01$ versus control group, &$P < 0.05$, &&$P < 0.01$ versus VPA group.

α diversity analysis indicated the significantly decreased Chao 1 index after VPA treatment and the alleviated change after its cotreatment with probiotics(Fig 4A). Differently, there was no significant differences in the levels of Shannon index among the different groups (Fig 4B). These findings reaveal that VPA treatment alters fecal bacterial community structure and lowers their richness, while probiotics treatment partially restores the alteration.

Next, the relative abundance of the top30 taxa at the phylum, family or genus level was shown in the boxplot (Fig 4D–4F). At the phylum level, the abundance of *Actinobacteriota*, *Acidobacteriota* and *Gemmatimonadota* was higher in VPA group than in control group, whereas probiotic treatment significantly reversed the changes (Fig 4G). At the family level, VPA group had a lower abundance of *Prevotellaceae*, *Christensenellaceae*, *Ruminococcaceae*, *[Eubacterium]_coprostanoligenes_group*, *Tannerellaceae* or *Butyricicoccaceael*, and a higher abundance of *Rs-E47-termite-group*, *Eggerthellaceae*, *Bukholderiaceae*, *Bifidobacteriaceae* or *Lactobacillaceae* than control group (Fig 4H). Nevertheless, probiotics treatment attenuated these changes with significant differences as for *Prevotellaceae* and *Christensenellaceae*. At the genus level, the relative abundance of *Prevotellaceae_UCG-001*, *Prevotellaceae_Ga6A1_group*, *Prevotella*, *Parabacteroides*, *Ruminococcus*, and *[Eubacterium]_coprostanoligenes_group* was significantly reduced upon VPA treatment, and the decreased *Prevotellaceae_UCG-001*, *Prevotellaceae_Ga6A1_group*, *Prevotella* were significantly prevented by probiotics (Fig 4I).

## Correlation between gut microbiota and indicators of liver steatosis or oxidative stress

The association between gut microbiota and liver steatosis- or oxidative stress- related indexes was clarified using the Spearman's rank correlation analysis. At the family level (Fig 5A), the TG level was negatively correlated with *Prevotellaceas*, *Oscillospiraceae*, *Clostridia_UCG-014*, *Tannerellaceae*, *UCG-010* and *Christensenellaceae* and positively correlated with *Muribaculaceae*, *Eggerthellaceae* and *Rs-E47_termite_group*. Also, *Desulfovibrionaceae* and *Erysipelatoclostridiaceae* were negatively associated with the TC level. Meanwhile, the MDA level had a

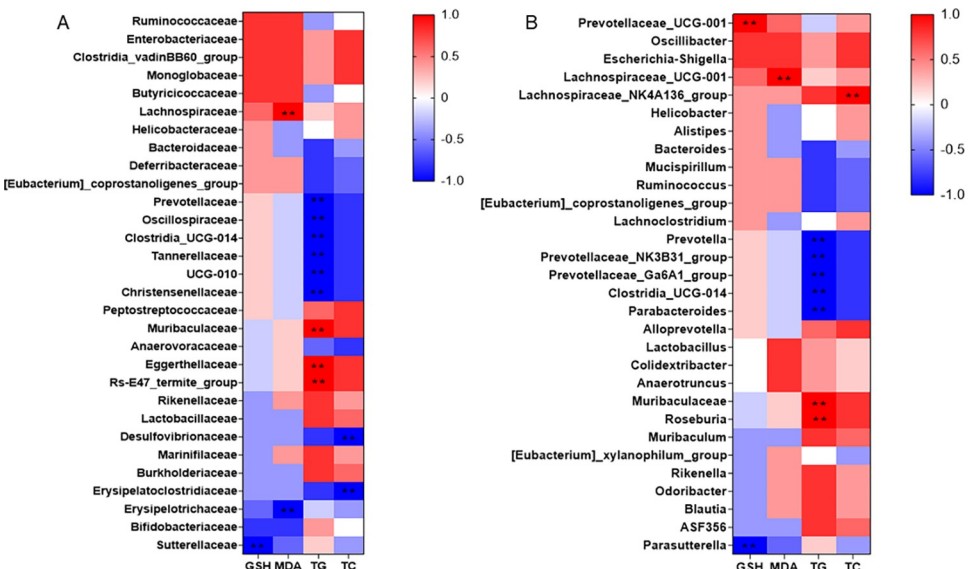

**Fig 5. Correlation between gut microbiota and indicators of liver steatosis or oxidative stress.** Heatmap of Spearman's correlation between the relative abundance of gut microbiota and liver steatosis- or oxidative stress-related indicators (A) at the family level and (B) at the genus level. Significant correlations are indicated by **$p < 0.01$.

significant positive and negative correlation with *Lachnospiraceae* and *Erysipelotrichaceae*, respectively. A negative association between *Sutterellaceae* and GSH level was also found.

At the genus level (Fig 5B), the TG level had a negative association with *Prevotella*, *Prevotellaceae_NK3B31_group*, *Prevotellaceae_Ga6A1_group*, *Clostridia_UCG-014* or *Parabacteroides*, and a positive association with *Muribaculaceae* and *Roseburia*. Further, a positive relatioiship between *Lachnospiraceae_NK4A136_group* and TC or *Lachnospiraceae_UCG-001* and MDA was shown. Additionally, the GSH level was positively and negatively correlated with *Prevotellaceae_UCG-001* and *Parasutterella*, respectively.

## Discussion

Probiotics have attracted interest in improving the intestinal bacterial composition and alleviating the development of NAFLD. Our study first demonstrated that a probiotic mixture attenuated VPA-induced liver steatosis and oxidative stress by manipulating gut microbiota, which provides further insight into the potential of probiotics in hepatic health and homeostasis.

Accumulating evidences have demonstrated that gut microbiota dysbiosis contributes to the initiation and progression of NAFLD [11]. Many preclinical and clinical studies have tried to investigate a microbial signature of NAFLD. Likewise, our results showed that VPA administration to mice promoted hepatic liver steatosis and oxidative stress accompanied with the altered structure and composition of gut microbiota at the level of phylum (increased *Actinobacteriota*, *Acidobacteriota* and *Gemmatimonadota*), family (decreased *Prevotellaceae*, *Christensenellaceae*, *Ruminococcaceae*, *[Eubacterium]_coprostanoligenes_group*, *Tannerellaceae*, or *Butyricicoccaceae* and increased *Rs-E47-termite-group*, *Eggerthellaceae*, *Bukholderiaceae*, *Bifidobacteriaceae* and *Lactobacillaceae*) or genera (decreased *Prevotellaceae_UCG-001*, *Prevotellaceae_Ga6A1_group*, *Prevotella*, *Parabacteroides*, *Ruminococcus*, and *[Eubacterium]_coprostanoligenes_group*). Interestingly, among these altered taxa, some of them have been demonstrated to have a relationship with NAFLD or obese. For example, the abundance of *Prevotellaceae UCG-001*, *Ruminococcus*, *Christensenellaceae*, and *[Eubacterium]_coprostanoligenes_group* was significantly decreased in mice or human fed with high-fat diet (HFD) [17–20], similar with our results in VPA-treated mice. Consistently, the reduced abundance of *Parabacteroides* was observed in patients with obesity or NAFLD [21–23]. Further, some studies demonstrated that patients with NAFLD had a consistent altered bacterial signature such as the decreased abundance of *Ruminococcaceae* family and *Prevotella* genera [24], which were also found in the study, indicating that there is an overlap of bacterial signatures between VPA-caused liver steatosis and high-fat diet-caused NAFLD. In addition, the heatmaps showed a significant correlation between some specific genera and family and the levels of TG, TC, GSH or MDA, suggesting that gut microbiota dysbiosis in VPA-treated mice are associated with liver steatosis and oxidative stress.

Moreover, a growing number of studies reveal that probiotics have a beneficial role in modulating gut microbiota [14]. Probiotics are considered to be a complementary approach for treating NAFLD. As expected, our results showed that probiotics treatment inhibited VPA-induced lipid accumulation with the significant restoration of gut microbiota including phylum of *Actinobacteriota*, *Acidobacteriota* and *Gemmatimonadota*, family of *Prevotellaceae* and *Christensenellaceae*, or genus of *Prevotellaceae_UCG-001*, *Prevotellaceae_Ga6A1_group*, *Parabacteroides* and *Prevotella*. Among them, *Prevotellaceae_UCG-001*, *Christensenellaceae*, *Prevotella*, and *Parabacteroides* have been demonstrated to have a close relationship with NAFLD. It suggests that probiotics-caused gut microbiota restortation contributes to the alleviation of liver steatosis in VPA-treated mice.

The present study found that VPA treatment not only promoted hepatic CYP2E1 overproduction but also reduced the levels and activities of antioxidant enzymes SOD, CAT, GST, and HO-1. CYP2E1 has been demonstrated to transform ethanol or polyunsaturated fatty acids into ROS and lipid peroxidation products [25, 26]. Meanwhile, antioxidant enzymes (SOD, CAT, GST, and HO-1) have a protective role against ROS accumulation [27]. Thereby, both the enhanced CYP2E1 generation and the inactivated antioxidant enzymes contribute to VPA-caused oxidative stress. Interestingly, the supplementation of probiotics significantly reversed these changes, resulting in the impaired oxidative stress. Similarly, some studies have demonstrated that probiotics exert their health benefits through the alleviation of oxidative stress [12]. It is reported that *Lactobacillus* and *Bifidobacterium* possess excellent antioxidant capacity by stimulating the activity of host antioxidant enzymes [12]. For example, *Lactobacillus plantarum* KSFY02 remarkably increased the enzymatic activities of CAT, SOD, and glutathione peroxidase in the liver of aged mice [28]. Treatment with a soy milk containing *Lactobacillus plantarum* A7 ameliorated oxidative stress in patients with diabetic kidney disease by elevating the levels of serum glutathione peroxidase and GSH [29]. The supplementation of probiotics containing *Bifidobacterium longum* CECT 7347, *Lactobacillus casei* CECT 9104, and *Lactobacillus rhamnosus* CECT 8361 enhanced ROS neutralization and plasma antioxidant levels in male cyclists [30]. Collectively, it is considered that the antioxidant capacity of *Lactobacillus* and *Bifidobacterium* is responsible for the reduced hepatic oxidative stress after probiotics treatment.

Also, we found that the mRNA and protein levels of nuclear Nrf2, a transcript factor of SOD, CAT, GST, and HO-1 were significantly upregulated after co-treatment with probiotics and VPA, suggesting Nrf2 has a positive role in the antioxidant capacity of probiotics, which has also been shown in two other studies [31–33]. One study exhibited the protective role of probiotics against oxidative stress by modulating the Nrf2-Keap1-ARE pathway [31]. The other study found that *Lactobacillus rhamnosus* GG treatment inhibited acetaminophen- or ethanol- caused acute oxidative liver damage in mice via upregulating the hepatic Nrf2 expression [33]. The molecular mechanism may be explained by the interaction of microbial metabolites with host receptors such as Toll-like receptors, which activates Nrf2-Keap1-ARE pathway and then promote the antioxidant enzyme expression.

In addition, VPA treatment caused liver steatosis accompanied with the upregulation of CD36 and DGAT2. CD36 is a transmembrane lipid transporter that promotes long-chain fatty acid uptake [34]. DGAT2 is a rate-limiting enzyme responsible for the transformation of diglycerides to triglycerides [35]. Thus, it suggests that the elevated CD36 and DGAT2 levels contribute to VPA-induced liver steatosis, which is further supported by the improved lipid deposit and the decreased CD36/ DGAT2 levels after probiotics treatment. Combined with previous research reavealing that oxidative stress is responsiple for the upregulation of CD36 and DGAT2 [36, 37], it is considered that probiotics could provide a certain degree of protection against VPA-caused liver steatosis through the alleviation of oxidative stress. Indeed, several preclinical studies have demonstrated that probiotics prevent NAFLD development by reducing oxidative stress [12, 38, 39]. Li and his colleagues found that *Lactobacillus plantarum* NCU116 played a beneficial role in NAFLD rats via the inhibition of oxidative stress and the restoration of gut microbiota [38]. Also, treatment with *Lactobacillus plantarum* NA136 to mice alleviated NAFLD by promoting antioxidant system and modulating fatty acid metabolism [39].

The 16s rRNA sequencing results showed that VPA treatment significantly lowered the abundance of several butyrating-producing bacteria such as *Prevotellaceae_UCG-001*, *Prevotellaceae_Ga6A1_group*, *[Eubacterium]_coprostanoligenes_group*, *Ruminococcaceae* or *Christensenellaceae* [40–46]. To our attention, previous studies showed that butyrate significantly

inhibited lipid deposit and oxidative stress in VPA-caused liver injury or insulin-resistant obese mice [47, 48]. We suppose that the butyrate might contribute to liver steatosis and oxidative stress in VPA-treated mice. Subsequently, we analyzed the short chain fatty acids (SCFAs) levels in the feces of mice by GC-MS. Surprisingly, there was no significant differences in the levels of butyric acid, valeric acid, propionic acid and acetic acid among the different groups S1 Fig, indicating that SCFAs might not be implicated in the alteration of hepatic fat accumulation and oxidative stress in the study. In addition, we also found that the mRNA levels of hepatic proinflammatory cytokines such as IL-6 and IL-1β were significantly downregulated in VPA group relative to control group S2 Fig, suggesting that VPA-caused liver steatosis and oxidative stress might not be associated with the inflammation. The molecular mechanism underlying the alleviation of VPA-caused liver steatosis and oxidative stress by probiotics remains to be verified in the future.

## Conclusions

The mixed probiotics intervention alleviated oxidative stress and lipid metabolic disorder in VPA-treated mice by inhibiting the pro-oxidant CYP2E1 level, activating the Nrf2/antioxidant enzyme pathway, and decreasing the CD36 and DGAT2 levels, accompanied with the restoration of gut microbiota dysbiosis. The data provide the evidence of probiotics efficacy and its potential utility in the prevention of liver steatosis.

## Supporting information

**S1 Fig. Effects of VPA and probiotics on the short chain fatty acids (SCFAs) levels in mice.** The levels of acetic acid (A), propionic acid (B), butyric acid (C), and valeric acid (D) in the feces of mice treated with vehicle, VPA, probiotics or in combination were analyzed by GC-MS method. Data were expressed as the MEAN±SD.
(TIF)

**S2 Fig. Effect of VPA on the expression of pro-inflammatory cytokines in mice.** The mRNA levels of IL-6 and IL-1β in the liver of mice treated with vehicle or VPA were analyzed by qRT-PCR method. Data were expressed as the MEAN±SD, $^{##}p < 0.01$ versus control group.
(TIF)

**S3 Fig. Raw data of all western blot images.**
(TIF)

**S1 Table. List of primers for qRT-PCR.**
(TIF)

## Acknowledgments

We thank Hong Zhang from the First Hospital of Jilin University for providing data analysis guidance in our experiment.

## Author Contributions

**Conceptualization:** Yingjie Guo.

**Formal analysis:** Xinrui Yan.

**Funding acquisition:** Yingjie Guo.

**Investigation:** Wenfang Song, Xinrui Yan, Yu Zhai, Jing Ren, Ting Wu, Han Guo, Yingjie Guo.

**Visualization:** Wenfang Song.

**Writing – original draft:** Yu Song, Xiaojiao Li, Yingjie Guo.

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
