## [Decision Letter · Decision Letter 0]

26 Jul 2023

PONE-D-23-14625Probiotics attenuate valproate-induced liver steatosis and oxidative stress in micePLOS ONE

Dear Dr. Guo,

Thank you for submitting your manuscript to PLOS ONE. After careful consideration, we feel that it has merit but does not fully meet PLOS ONE’s publication criteria as it currently stands. Therefore, we invite you to submit a revised version of the manuscript that addresses the points raised during the review process.

We look forward to receiving your revised manuscript.

Kind regards,

Juan J Loor

Academic Editor

PLOS ONE

Journal Requirements:

3. To comply with PLOS ONE submissions requirements, in your Methods section, please provide additional information regarding the experiments involving animals and ensure you have included details on (1) methods of sacrifice, (2) methods of anesthesia and/or analgesia, and (3) efforts to alleviate suffering.

4. We suggest you thoroughly copyedit your manuscript for language usage, spelling, and grammar. If you do not know anyone who can help you do this, you may wish to consider employing a professional scientific editing service. 

Reviewers' comments:

Reviewer's Responses to Questions

**Comments to the Author**

1. Is the manuscript technically sound, and do the data support the conclusions?

Reviewer #1: Yes

2. Has the statistical analysis been performed appropriately and rigorously? 

Reviewer #1: No

3. Have the authors made all data underlying the findings in their manuscript fully available?

Reviewer #1: Yes

4. Is the manuscript presented in an intelligible fashion and written in standard English?

Reviewer #1: Yes

5. Review Comments to the Author

Reviewer #1: The subject is very interesting and useful and the article is very well written.

Please mention the weight of the mice.

Please depict a graphical abstract including results.

For statistical analysis of the body weight, repeated measure ANOVA is needed. Please correct it and mention it in the statistical analysis.

The experimental study included four groups, and one-way ANOVA is the correct analysis. When do the authors use T student?

6. PLOS authors have the option to publish the peer review history of their article (what does this mean?). If published, this will include your full peer review and any attached files.

Reviewer #1: No

---

## [Author Response · Author response to Decision Letter 0]

23 Aug 2023

Dear Dr. Loor，

We sincerely thank you and the reviewers for the helpful comments on our manuscript (PONE-D-23-14625). Those comments are all valuable and very helpful for revising and improving our paper. We have revised the text in order to address the reviewers’ comments. The specific changes have been indicated using the Track Changes feature.

Journal Requirements:

Response: Thank you for your reminder. We have ensured the manuscriot meets PLOS ONE’s style requierments.

Response:Thank you for you reminder. We have provided the original underlying images for all immunoblotting data reported in the figure2 B, D and F, and Figure 3C , and uploaded the file named as “S1 Raw images” in the supporting information.

3. To comply with PLOS ONE submissions requirements, in your Methods section, please provide additional information regarding the experiments involving animals and ensure you have included details on (1) methods of sacrifice, (2) methods of anesthesia and/or analgesia, and (3) efforts to alleviate suffering.

Response: Thank you for you suggestions. We have provided the information regarding the experiments involving animals (1) methods of sacrifice and (2) methods of anesthesia and/or analgesia,(3) efforts to alleviate suffering in the methods section of revised manuscript. 

4. We suggest you thoroughly copyedit your manuscript for language usage, spelling, and grammar. If you do not know anyone who can help you do this, you may wish to consider employing a professional scientific editing service. 

Response:Thank you for you suggestions. The whole manuscript for language usage, spelling, and grammar using track changes was carefully revised by Shuang Wu, who graduated from School of foreign language, Jinan University. The revised manuscriot with track changes was uploaded as a separate file “Manuscript revised language with track changes” in supporting information. 

5.PLOS requires an ORCID iD for the corresponding author in Editorial Manager on papers submitted after December 6th, 2016. Please ensure that you have an ORCID iD and that it is validated in Editorial Manager. To do this, go to ‘Update my Information’ (in the upper left-hand corner of the main menu), and click on the Fetch/Validate link next to the ORCID field. This will take you to the ORCID site and allow you to create a new iD or authenticate a pre-existing iD in Editorial Manager. Please see the following video for instructions on linking an ORCID iD to your Editorial Manager account: https://www.youtube.com/watch?v=_xcclfuvtxQ

Response: Thank you for you suggestions. We have ensured the corresponding author’ ORCID iD 0000-0003-4203-8296 validated in Editorial Manager. 

Response: Thank you for you suggestions. We have added the Figure S1 and Figure S2 in the supporting information, which show the finding of “data not shown” in the discussion section of manuscript. In the revised manuscript, we have substituted the phase “data not shown” with Figure S1 and Figure S2, which are marked with red words of revised manuscript. 

7.Please include your full ethics statement in the ‘Methods’ section of your manuscript file. In your statement, please include the full name of the IRB or ethics committee who approved or waived your study, as well as whether or not you obtained informed written or verbal consent. If consent was waived for your study, please include this information in your statement as well.

Response: Thank you for you suggestions. We have added the full name of the ethics committee in the Methods section, and all of us obtained informed written consent, as follows “ All animal protocols and experiments were approved by the Animal Ethics Committee of Jilin University, in accordance with the Guidelines for the Care and Use of Laboratory Animals of Jilin University (Protocol Number: 2021-PZ012). Everyone obtained informed written consent”, which were marked with red words in the Methods section of revised manuscript. 

Reviewers' comments:

Reviewer's Responses to Questions

Comments to the Author

1. Is the manuscript technically sound, and do the data support the conclusions?

Reviewer #1: Yes

2. Has the statistical analysis been performed appropriately and rigorously?

Reviewer #1: No

3. Have the authors made all data underlying the findings in their manuscript fully available?

Reviewer #1: Yes

4. Is the manuscript presented in an intelligible fashion and written in standard English?

Reviewer #1: Yes

5. Review Comments to the Author

Reviewer #1: The subject is very interesting and useful and the article is very well written.

Please mention the weight of the mice.

Response: Thank you for you suggestions. We have added the description of the weight of the mice “20-22 g” in the section of Animals and treatment of manuscript, which are marked with red words. 

Please depict a graphical abstract including results.

Response: Thank you for you suggestions. We have added the description of the graphical abstract, as shown below.

“ VPA administration to mice enhanced the hepatic oxidative stress and lipid accumulation by increasing the pro-oxidant CYP2E1 level, inhibiting the Nrf2/antioxidant enzyme pathway, and upregulating the CD36 and DGAT2 levels, accompanied with the gut microbiota dysbiosis, all of which were significantly alleviated by co-treatment with probiotics.”

For statistical analysis of the body weight, repeated measure ANOVA is needed. Please correct it and mention it in the statistical analysis.

Response: Thank you for your suggestions. For statistical analysis of the body weight, multiple comparision between groups at every time point were performed, but the comparison of body weight within group between different time points was not adopted in the study. Therefore, one way ANOVA with multiple comparison, but not repeated measure was used for the statistical analysis of the body weight in the study. In addition, for clear view of significant differences of the body weight between groups, we have replaced the original line chart with a bar chart, and the body weight were recorded every six days, not every three days, as shown in the revised Figure 1A.

The experimental study included four groups, and one-way ANOVA is the correct analysis. When do the authors use T student?

Response: Thank you for your good suggestions. As you described, one-way ANOVA analysis was carried out for all the four groups involved in the experimental results, which have been carefully re-analyzed and shown in revised figures. T-test for comparison between the two groups was involved in Figure 3A. The statistical anaylsis in detail was described in the section of Statistical analysis in the revised manuscript.

Your Sincerely

Yingjie Guo

School of Life Sciences, Jilin University

---

## [Decision Letter · Decision Letter 1]

11 Oct 2023

PONE-D-23-14625R1Probiotics attenuate valproate-induced liver steatosis and oxidative stress in micePLOS ONE

Dear Dr. Guo,

Thank you for submitting your manuscript to PLOS ONE. After careful consideration, we feel that it has merit but does not fully meet PLOS ONE’s publication criteria as it currently stands. Therefore, we invite you to submit a revised version of the manuscript that addresses the points raised during the review process.

MY APOLOGIES FOR THE BACK AND FORTH ON THE PAPER. THERE WAS ONE FINAL SUGGESTION FROM THE REVIEWER THAT I BELIEVE IS APPROPRIATE. PLEASE CONSIDER IT AND IF YOU ELECT NOT TO MAKE THE REVISION INDICATE IN THE STATISTICAL ANALYSIS SECTION OF THE PAPER THE REASON WHY A REPEATED MEASURES ANALYSIS WAS NOT USED.

We look forward to receiving your revised manuscript.

Kind regards,

Juan J Loor

Academic Editor

PLOS ONE

Journal Requirements:

Reviewers' comments:

Reviewer's Responses to Questions

**Comments to the Author**

1. If the authors have adequately addressed your comments raised in a previous round of review and you feel that this manuscript is now acceptable for publication, you may indicate that here to bypass the “Comments to the Author” section, enter your conflict of interest statement in the “Confidential to Editor” section, and submit your "Accept" recommendation.

Reviewer #1: All comments have been addressed

2. Is the manuscript technically sound, and do the data support the conclusions?

Reviewer #1: Yes

3. Has the statistical analysis been performed appropriately and rigorously? 

Reviewer #1: No

4. Have the authors made all data underlying the findings in their manuscript fully available?

Reviewer #1: Yes

5. Is the manuscript presented in an intelligible fashion and written in standard English?

Reviewer #1: Yes

6. Review Comments to the Author

Reviewer #1: All comments were addressed by the authors. For analyzing the the body weight, repeated measures should be done.

7. PLOS authors have the option to publish the peer review history of their article (what does this mean?). If published, this will include your full peer review and any attached files.

Reviewer #1: **Yes: **Hamideh Bashiri

---

## [Author Response · Author response to Decision Letter 1]

15 Oct 2023

Dear Dr. Loor， 

We sincerely thank you and the reviewers for the helpful comments on our manuscript (PONE-D-23-14625R1). Those comments are all valuable and very helpful for revising and improving our paper. We have revised the text in order to address the reviewers’ comments. The specific changes have been indicated using the Track Changes feature. 

Journal Requirements:

Response: Thank you for your reminder. We have ensured that all listed references are complete and correct in the manuscript. Meanwhile, there are not cited papers retracted

Reviewers' comments:

Reviewer's Responses to Questions

Comments to the Author

1. If the authors have adequately addressed your comments raised in a previous round of review and you feel that this manuscript is now acceptable for publication, you may indicate that here to bypass the “Comments to the Author” section, enter your conflict of interest statement in the “Confidential to Editor” section, and submit your "Accept" recommendation.

Reviewer #1: All comments have been addressed

2. Is the manuscript technically sound, and do the data support the conclusions?

Reviewer #1: Yes

3. Has the statistical analysis been performed appropriately and rigorously?

Reviewer #1: No

Response: Thank you for your suggestions. For statistical analysis of the body weight, comparisons between time-based measurements within each group were performed with analysis of variance for repeated measurement ANOVA, and a P value <0.05 was considered to be statistically significant, as indicated in the section of 2.8 Statistical analysis and the revised Figure 1A.

4. Have the authors made all data underlying the findings in their manuscript fully available?

Reviewer #1: Yes

5. Is the manuscript presented in an intelligible fashion and written in standard English?

Reviewer #1: Yes

6. Review Comments to the Author

Reviewer #1: All comments were addressed by the authors. For analyzing the the body weight, repeated measures should be done.

Response: Thank you for your suggestions. For statistical analysis of the body weight, comparisons between time-based measurements within each group were performed with analysis of variance for repeated measurement ANOVA, and a P value <0.05 was considered to be statistically significant, as indicated in the section of 2.8 Statistical analysis and the revised Figure 1A.

7. PLOS authors have the option to publish the peer review history of their article (what does this mean?). If published, this will include your full peer review and any attached files.

Do you want your identity to be public for this peer review? For information about this choice, including consent withdrawal, please see our Privacy Policy.

Reviewer #1: Yes: Hamideh Bashiri

---

## [Editor Report · Decision Letter 2]

31 Oct 2023

Probiotics attenuate valproate-induced liver steatosis and oxidative stress in mice

PONE-D-23-14625R2

Dear Dr. Guo,

We’re pleased to inform you that your manuscript has been judged scientifically suitable for publication and will be formally accepted for publication once it meets all outstanding technical requirements.

Kind regards,

Juan J Loor

Academic Editor

PLOS ONE
---

## [Editor Report · Acceptance letter]

7 Nov 2023

PONE-D-23-14625R2 

Probiotics attenuate valproate-induced liver steatosis and oxidative stress in mice 

Dear Dr. Guo:

I'm pleased to inform you that your manuscript has been deemed suitable for publication in PLOS ONE. Congratulations! Your manuscript is now with our production department. 

Kind regards, 

on behalf of

Dr. Juan J Loor 

Academic Editor

PLOS ONE